# Electrochemical Biosensors for Circulating Tumor DNA Detection

**DOI:** 10.3390/bios12080649

**Published:** 2022-08-17

**Authors:** Ke Wang, Zhijia Peng, Xiaogang Lin, Weiqi Nian, Xiaodong Zheng, Jayne Wu

**Affiliations:** 1Key Laboratory of Optoelectronic Technology and Systems of Ministry of Education of China, Chongqing University, Chongqing 400044, China; 2Chongqing University Cancer Hospital, Chongqing University, Chongqing 400044, China; 3Department of Electrical Engineering and Computer Science, University of Tennessee, Knoxville, TN 37996, USA

**Keywords:** non-destructive, biosensors, real-time detection, circulating tumor DNA (ctDNA), high sensitivity, Internet of Things

## Abstract

Early diagnosis and treatment have always been highly desired in the fight against cancer, and detection of circulating tumor DNA (ctDNA) has recently been touted as highly promising for early cancer-screening. Consequently, the detection of ctDNA in liquid biopsy is gaining much attention in the field of tumor diagnosis and treatment, which has also attracted research interest from industry. However, it is difficult to achieve low-cost, real-time, and portable measurement of ctDNA in traditional gene-detection technology. Electrochemical biosensors have become a highly promising solution to ctDNA detection due to their unique advantages such as high sensitivity, high specificity, low cost, and good portability. Therefore, this review aims to discuss the latest developments in biosensors for minimally invasive, rapid, and real-time ctDNA detection. Various ctDNA sensors are reviewed with respect to their choices of receptor probes, designs of electrodes, detection strategies, preparation of samples, and figures of merit, sorted by type of electrode surface recognition elements. The development of biosensors for the Internet of Things, point-of-care testing, big data, and big health is analyzed, with a focus on their portable, real-time, and non-destructive characteristics.

## 1. Introduction

From the development of the first biosensor—a glucose sensor—to the present, biosensor technology has become an interdisciplinary field that combines biology, chemistry, physics, medicine, electronic technology, and other disciplines [1]. In recent years, biosensors have become increasingly important candidates for early cancer-screening and tumor-marker detection because of their small size, lower sample volume, short detection time, high sensitivity and accuracy, and lesser damage to organisms [2]. In the detection of tumors, traditional biopsy requires a fragment of the tumor, cannot be performed frequently, and can be rather invasive to the patient. Therefore, to make early detection of cancer possible during a routine checkup, there is an urgent need for a minimally invasive, low-cost, and near-real-time detection technology [3].

In recent years, circulating tumor DNA (ctDNA) detection technology has been widely studied; it was rated as one of the top ten breakthrough technologies in 2015 by MIT Magazine [4]. The family of ctDNA molecules is a mixture of single- and double-stranded DNAs of around 166 nucleotides that are released as free DNA by circulating tumor cells after necrosis and apoptosis. They exist in the peripheral blood and can potentially act as a diagnostic biomarker with high specificity and sensitivity [5]. General ctDNA detection techniques can be divided into digital PCR [6], droplet digital PCR [7], magnetic bead emulsion amplification [8], label amplification depth sequencing [9], etc. However, these technologies generally suffer from the disadvantages of high cost, poor portability, frequent occurrence of false positives, and long assay time, which are not suitable for on-site detection and mobile medical treatment. In contrast, ctDNA electrochemical biosensor technology has high potential to be implemented for field-deployable detection, because of its specificity, portability, high sensitivity, and fast response [10].

As shown in Figure 1, a ctDNA electrochemical biosensor is a device that converts the biological signal generated by specific binding between the recognition probe with the target ctDNA into an electrical signal for detection. Electrochemical biosensors can analyze biological information from the impedance of multiple frequencies or one frequency band [11,12]. In addition, their unique advantages of low cost, portability, and near-real-time detection make them suitable for on-site screening [13] and mobile health monitoring and could become an enabling tool for personalized medicine in the context of big data, the Internet of Things, and big health [14,15].

This paper reviews the research progress of ctDNA electrochemical biosensors in recent years, focusing on the types of recognition elements on the electrode surface of ctDNA electrochemical biosensors. The organization is as follows: Section 2 describes the categories of recognition elements for ctDNA electrochemical biosensors; Section 3 and Section 4 summarize the research progress of ctDNA electrochemical biosensors in recent years; Section 5 discusses other types of ctDNA biosensors; and Section 6 analyzes the prospects and challenges in the development of ctDNA electrochemical biosensors. Finally, the characteristics of ctDNA biosensors are summarized.

## 2. Structure of Surface-Based Biosensors

Electrochemical biosensors generally consist of three main parts: the first part comprises the biomolecular recognition elements, as shown in Table 1, which have the ability to recognize the target analyte with high specificity [16]. The biomolecular recognition elements of ctDNA electrochemical biosensors can be categorized into enzymes, nucleic acids, organelles, immune substances, etc. [11]. Biomolecular receptors can be fixed on the surface of a biosensor by physical adsorption [17], covalent bonding [18,19,20], embedding [21,22,23], self-assembled membrane [24,25,26], affinity [27,28], and other methods, as shown in Figure 2. The second part consists of conductive electrodes, which are widely used as signal conversion elements [29]. DNA, antibodies, and other recognition elements can be fixed stably on the electrode, while maintaining their activity, to realize the function of a biosensor. Common electrode types include gold electrodes, carbon electrodes, and graphene electrodes. They can convert biochemical reactions into electrical signals, which may be further amplified and processed to yield the concentrations of the target analytes. Therefore, a variety of surface modification techniques and different transducers can be used to construct ctDNA detection techniques. The third part is the signal transduction mechanism, which uses the basic principles and experimental techniques of electrochemistry to determine the composition and content of substances according to their electrochemical properties. The commonly used methods of electrical analysis include cyclic voltammetry (CV), square wave voltammetry (SWV), differential pulse voltammetry (DPV), and electrochemical impedance spectroscopy (EIS).

At present, clinical detection of ctDNA is mainly based on polymerase chain reaction (PCR) technology and gene sequencing, while there are few reports on ctDNA biosensors, partly due to the small sizes of ctDNA. Compared with PCR and gene sequencing, ctDNA biosensors are of lower cost and higher sensitivity, and are easier to integrate and carry, making them more suitable for on-site detection and mobile medical development.

In the following, different biosensing approaches applied to ctDNA detection are discussed based on the existing ctDNA biosensor reports.

## 3. Nucleic Acid Probe-Based Detection

Nucleic acid identification biosensors are widely used in clinical diagnosis [30,31], microbial detection [32], and environmental monitoring [33] because of their high sensitivity, fast response, simple operation, low price, and miniaturization. The probe can recognize the target DNA molecule by specific hybridization and convert the reaction into an electrical signal for analysis.

### 3.1. PNA Probe

At present, DNA sensors mainly use peptide nucleic acid (PNA) as a probe to specifically bind ctDNA to capture the target DNA. Since there is no electrostatic repulsion between PNA and DNA, the ability of PNA to hybridize with DNA molecules is superior to DNA/DNA. The PNA probe is fixed on the surface of the sensor using covalent bond modification technology, that is, the sulfhydryl group of PNA forms a Au–S bond with the gold electrode, so that the PNA is immobilized on the electrode surface. Cai et al. [34] proposed a dual biomarker detection platform based on a PNA probe fixed with gold nanoparticles and lead phosphate apoferritin for the detection of PIK3CA gene tumor characteristic mutation and methylation in order to quantify ctDNA. DNA methylation refers to the DNA methylation transfer enzyme under the action of the genome CpG dinucleotide cytosine 5 carbon covalent bond combined with a methyl group. The results were quantified using SWV detection by an electrochemical workstation.

First, a large amount of PNA was fixed on the surface of a screen-printed electrode (SPE) with gold nanoparticles as the carrier through covalent bonds. The SPE electrode consists of a carbon working electrode, a carbon auxiliary electrode, and an Ag/AgCl reference electrode. PNA probes can form stable complexes with DNA through complementary base-pairing or Hoogsteen base-pairing principle [35,36]. Serum samples were collected from ten cancer patients, and four of them were randomly selected for the experiment. When a solution of ctDNA with mutant sites was dropped onto the SPE’s working electrode, the mutant sites bound to the PNA probe in a complementary and specific way. Afterwards, monoclonal antibody carrier lead phosphate apoferritin (LPA) solution was added, which could amplify the selectivity of ctDNA, thus enhancing the electrochemical detection signal. In contrast, when normal ctDNA (ncDNA) solution was dropped to the electrode, ncDNA could neither bind complementary to the PNA probe nor bind to mAb-LPA, resulting in almost no potential change. Compared to normal cells, the genetic changes after cell carcinoma were increase of the methylation level of the anti-cancer gene and decrease of the methylation level of the primary cancer gene. Therefore, abnormal levels of ctDNA methylation also can be used to verify whether a tumor is present in the body. This new method based on DNA probe complementation had excellent selectivity and ultra-high sensitivity, and the detection limit was 1.0 × 10^−14^ mol/L.

To further improve the selectivity of a PNA-probe-based ctDNA sensor, Das et al. [37] designed a DNA clutch probe to prohibit the recombination of ssDNA stands. The DNA double-helix structure denatured at 90 °C to form ssDNA, and then DNA clutch probes were added to prevent ssDNA chains from rebinding. Clamp for wild-type was added to bind to the wild-type target ssDNA and keep the mutant target ctDNA still single-stranded. PNA probes on nanostructured microelectrodes bound to mutant target DNA. Finally, the signal generated from individual sensors was measured in the presence of an electrocatalytic reporter system using differential pulse voltammetry. The PNA probe was fixed on the gold electrode by covalent bonding method, which specifically bound the mutated target gene ssDNA, resulting in the change of electrode potential. The sensor successfully detected mutated ctDNA in samples from lung cancer and melanoma patients. Nguyen et al. [38] captured and enriched 69 bp PIK3CA ctDNA using peptide nucleic acid (PNA) as a probe. The ultra-sensitive detection of tumor-specific mutations (E542K and E545K) and ctDNA methylation of PIK3CA gene were performed using a coupled plasma model based on local surface plasmon resonance (LSPR) and gold nanoparticles (AuNPs). The integrated biosensor system included a dark field microscope, spectrometer, and CCD camera for illumination, acquisition, and Rayleigh scattering spectrum signal conversion. Immunogold colloid was used as an enhanced secondary reaction based on the methylation detector and plasma coupling to detect and amplify methylation by specifically binding immunogold colloid to CpG sites of methylation on ctDNA sequence. To mimic clinical samples, 50 fM ctDNA of E542K and E545K was added to commercial human serum (H4522, Sigma-Aldrich, USA), and the mixture was injected into a biosensor, which was controlled at 62 °C. When ctDNA molecules bound to the PNA probe, the refractive index (RI) of the biosensor surface changed, resulting in an obvious LSPR peak shift. The concentration of ctDNA was determined according to the peak shift distance. The limit of detection (LOD) of ctDNA was 50 fM in the range of 50–3200 fM, corresponding to the LSPR peak offset of 4.3 nm.

### 3.2. DNA Probe

For the detection of ctDNA, Mahbubur et al. [39] modified graphene-oxide-coated gold nanoparticles onto the glassy carbon electrode and fixed the target probe through the π–π interaction between DNA bases to detect the ctDNA of the PIK3CA gene in the peripheral blood of gastric cancer. The hybridization of the probe DNA with ctDNA caused the resulting dsDNA to detach from the electrode’s surface, leading to an increase of electrical current. The detection limit was as low as 1.0 × 10^−20^ mol/L. This method can be used to analyze ctDNA in serum samples of cancer patients, which has great potential for the application of ctDNA detection in real-time detection research.

Zhang et al. [40] established a MoS_2_ nanosheet polymer biosensor based on poly-xanthurenic acid film functionalization. As described in Figure 3, these polymer biosensors were electropolymerized by poly-xanthurenic acid (PXA) on the surface of a MoS_2_ electrode that was prepared in advance, thus forming an ideal interface for the PIK3CA gene expressed in the peripheral blood of patients with gastric cancer. Then, the probe ssDNA was directly fixed on the PXA/MoS_2_ nanocomposite, and the change of self-signal after ssDNA hybridization with target DNA was induced by cyclic voltammetry and electrochemical impedance spectroscopy. The limit of detection of the polymer biosensor was reported to be 1.8 × 10^−17^ mol/L for the PIK3CA gene [40].

Zhao et al. [41] prepared nanocomposite MWCNT–PDA–Au–Pt by uniformly dispersing Au–Pt alloy nanoparticles on MWCNT–PDA. The nanocomposite MWCNT–PDA–Au–Pt had a good reduction effect on H_2_O_2_ and could amplify the current response. It could be mixed with signal probe (SPs) and 6-mercapto-1-hexanol (MCH) to form SPs-label, which was used to construct the electrochemical biosensor. The capture probe (CPs) was fixed on the surface of the screen-printed gold electrode (SPGE) by Au–S bond. Both SPs and CPs are DNA. CPs recognized and captured the target ctDNA. Then, SPs-label was added in order to form a sandwich structure by base-pairing (See Figure 4). The ctDNA samples have been linked to triple-gene-negative breast cancer. When the target ctDNA was detected, the current characteristics changed significantly, and the corresponding current was lower than that when ctDNA was not detected. The linear detection range of the ctDNA biosensor was from 1 × 10^−15^ mol/L to 1 × 10^−8^ mol/L, and the detection limit was as low as 5 × 10^−16^ mol/L.

In 2018, Wang et al. [42] designed a multifunctional label-free circulating tumor DNA KRAS G12DM enzyme electrode biosensor based on the terminal deoxynucleotide transferase (TdT) and ribonucleic acid HII (RNase HII) dual-enzyme co-group amplification strategy. As shown in Figure 5, the biosensor consisted of a triple-helix molecular switch (THMS) as molecular recognition and signal transduction probe and ribonuclease HII (RNase HII) and terminal deoxynucleotidyl transferase (TdT) as dual enzyme assisted multiple amplification accelerator. Signal transduction probe (STP) was released under the action of RNase HII, and the capture probe fixed on the gold electrode was hybridized with STP, and then, TdT was used to achieve TdT-mediated cascade expansion to generate a stable DNA dendritic structure. Finally, the electrically active molecule methylene blue (MB) was obtained. Ten clinical plasma samples were tested, including five from colorectal cancer patients and five healthy donors. The labeled electroactive tree achieved highly sensitive and accurate detection of ctDNA. The limit of detection was down to aM.

In order to improve the detection accuracy of ctDNA by sensor, Peng et al. [43] generated sea-urchin-shaped gold nanocrystals (U–Au) by the synergies of CTAC, KBr, KI, and L-glutathione. U–Au-modified multigraphene aerogel was prepared on a glass carbon electrode (GCE). The aim was to achieve cyclic amplification induced by target DNA. DNA probe 1 (P1) with methylene blue (MB) was hybridized with DNA probe 2 (P2) with ferrocene (Fc) to form double-stranded DNA, which was linked to U–Au by Au–S bond. Catalytic REDOX of Fc and MB enhanced the detection signal. This biosensor was used to detect the ctDNA of KRAS gene associated with colorectal cancer in diluted serum samples. In the presence of target DNA, ctDNA complementary paired with Hairpin DNA 1 (H1) on the electrode surface through base-pairing, forming an open hairpin structure of H1, triggering the target DNA-induced cycle, making more MB probes move closer to the electrode surface and more Fc probes move away from the electrode surface, resulting in an increase in the peak current of MB. The peak current of Fc was reduced, thus realizing the detection of ctDNA. The electrochemical response increased with ctDNA concentration from 0.1 fM to 1 × 10^6^ fM, and the detection limit was reported to be 0.033 fM.

Li et al. [44] made silicon nanowire (SiNW) array sensors on silicon-on-insulator (SOI). The preparation process of the sensor is shown in Figure 6. First, the entire surface of the sensor was treated with oxygen (O_2_) plasma and then immersed overnight in APTES and 99.5% ethanol at room temperature. The carboxyl groups at the end of the probe DNA were activated by a mixture of NHS and EDC (1:1) prior to immobilizing probe DNA. Subsequently, the specific single-stranded DNA (ssDNA) probe of PBS was assembled on the surface of SiNW, and the recognition functional layer (ssDNA/SINW-array FET) was formed on the surface of the sensor. For the biosensing experiments, blood samples were collected from ten healthy volunteers and five of them were randomly selected. Serum samples were separated by centrifugation of the supernatant at 3000 rpm for 15 min. Finally, PIK3CA E542K ctDNA solution was pipetted into the PDMS well for specific recognition and detection of ctDNA. The experimental results showed that the biosensor has obvious advantages in ctDNA detection. The detection range of ctDNA concentration was 0.1 Fm–100 pM, and the ultra-low detection limit was reported to be 10 aM, with good linearity.

In 2022, Miao et al. [45] proposed the concept of DNA nanostructure transformation and used the principle of DNA bipedal walker and base complementary pairing based on DNA nanostructure transformation to specifically recognize ctDNA. First, a DNA walking track was constructed on the surface of the gold electrode. Cytosine-rich probe A was fixed on the electrode through Au–S chemical bond, and probe B was connected through Hoogsteen reaction. Alkaline conditions could destroy the reaction, release probe B, and allow the electrode to regenerate and realize recycling. The target opened probe E to expose the template sequence, which was polymerized under the action of polymerase to form the double chain containing the cutting site, and then generated the fragment with the same sequence as the target under the action of the incision enzyme. A large number of single chains were obtained under multiple cycles. Dumbbell-shaped probe D released two rings under the action of the single-chain product, and the 5′ and 3′ ends of probe D combined to form a wheel shape. DNA enzymes on the wheel-shaped DNA encouraged the walker to walk, the track on the electrode surface was cut, and the electrochemically active substance disengaged from the electrode, causing the signal to weaken. Continuous SDA and walking nanomachines could achieve double amplification, ensuring the ultra-high sensitivity of ctDNA detection. Clinical samples were collected from both healthy individuals and breast cancer patients. After peripheral blood samples were collected and centrifuged, the serum was diluted 10-fold with 10 mM Tris-HCl. For the detection experiments, the DNA assembly on the electrode surface was characterized by chronocoulometry (CC), EIS, CV, and SWV. The concentration of ctDNA was calculated using the measured peak current of SWV and relevant formulas. The detection limit of this method was reported to be as low as 2.2 aM, and the specificity was strong.

Liu et al. [46] developed a rapid, accurate, and cost-effective assay for the detection of ctDNA EGFR L858R based on the CRISPR/Cas12a system and MB/Fe_3_O_4_@COF/PdAu nanocomposites. The schematic of this approach is shown in Figure 7. The CRISPR/Cas12a system has played an increasingly important role in the precise identification of ctDNA targets and cuts of single-stranded DNA. MB/Fe_3_O_4_@COF/PdAu nanocomposites had good catalytic activity and signal amplification performance. In the absence of target DNA, the trans-cleavage activity of Cas12a was silenced and the single-stranded DNA remains intact. Thus, the immobilized CP1 and MB/Fe_3_O_4_@COF/PdAu/CP2 nanocomplexes could bind with single-stranded DNA by complementing 5′ and 3′ single-stranded DNA, respectively, resulting in relatively high MB signals. When the target was present, CP1 and CP2 could not be linked by single-stranded DNA, because Cas12a trans-cleavage activity was activated to cleave nonspecific single-stranded DNA. Therefore, the MB signal was reduced relative to the background. Therefore, ctDNA EGFR L858R could be quantitatively detected according to the change of current. The linear range of the electrochemical biosensor was 10 aM–100 pM, and the detection limit was 3.3 aM. This sensor provides a high-precision, reliable, and convenient ctDNA detection method, which has good potential in the diagnosis and prognosis of cancer.

Using a 3D graphene-like homogeneous carbon structure (3D-GHC_600_) loaded with gold platinum (AuPt), Chen et al. [47] developed a ctDNA electrochemical biosensor with a low detection limit and high selectivity. The synthesis process of AuPt/3D-GHC_600_ composite catalyst and the preparation steps of ctDNA electrochemical biosensor are shown in Figure 8. The advantages of 3D-GHC_600_ include large area, abundant mesoporous nature, uniform size and morphology, and 3D structure. AuPt was formed on the 3D-GHC_600_ surface by in situ reduction reaction. In addition, the catalyst was used as a tag for signal probes (SPs). The hybridization reaction is accomplished by layer-by-layer recognition of captured probe (CPs), target DNA (ctDNA), and SPs markers on the electrode, resulting in sandwich-like structures. In ctDNA detection analysis, the current signal from the electrocatalytic reduction of H_2_O_2_ by SPs labeling was recorded. The ctDNA biosensor showed good performance, including a linear range of 10^−8^ M–10^−17^ M and a detection limit of 2.25 × 10^−18^ M. The sensor also showed excellent selectivity, satisfactory reproducibility, good stability, and excellent recovery.

### 3.3. RNA Probe

Uygun et al. [48], for the first time, modified the inactivated Cas9 (dCas9) protein and synthetic guide RNA (sgRNA) on a graphene oxide screen-printed electrode (GPHOXE) as a biometric receptor for labeling free detection of circulating tumor DNA (ctDNA). This was achieved by sequence-specific recognition and EIS analysis to detect tumor-associated mutations (PIK3CA exon 9 mutations); the biosensor modified by dCas9–sgRNA is shown in Figure 9. Covalently modifying the activated electrode with dCas9 reduced the impedance curve due to the reduced number of freely available carboxyl groups, which kept the REDOX probe away from the surface, while the positively charged dCas9 protein attracted the REDOX probe, which reduced the electron transfer resistance. On the contrary, when sgRNA modification and ctDNA were successfully combined, electron transfer resistance increased significantly. The dCas9–sgRNA modified biosensor had a linear detection range of 2–20 nM for 120 bp ctDNA within 40 s. The calibration curve showed good linearity, with the lowest detection limit (LOD) calculated at 0.65 nM and the lowest quantification limit (LOQ) calculated at 1.92 nM.

## 4. Antibody Probe-Based Detection

Antibodies have been widely used in disease diagnosis and treatment, food safety control, and environmental monitoring [49]. Specific binding by an antibody is the foundation for the detection of pathogens, micro-molecules, cells, bacteria, and other molecules with biological characteristics. Detection based on antibody probe has the advantages of reducing non-specific interference and lowering the lower limit of detection. Antibody specific to target ctDNA is immobilized on the electrodes. Captured ctDNA was detected by electrochemical method or enzyme-linked immunosorbent assay.

At present, DNA site-specific methylation has been increasingly used as a biomarker [50,51,52]. In addition, ctDNA methylation is an important epigenetic modification to control tumors, and detection of ctDNA methylation level can effectively determine the degree of malignancy of tumors [53]. The methods for analyzing ctDNA methylation mainly include PCR, sequencing, microarray, etc., but all these methods require ctDNA pretreatment [54,55]. In contrast, 5-methylcytosinine (5-mC) monoantibodies can be directly immobilized on the electrode by covalently coupling method, which can be used to capture methylated ctDNA without sample pretreatment. In 2018, Povedano et al. [56] proposed to use a 5-mC single antibody as a ctDNA receptor and used a hydrogen peroxide/hydroquinone (H_2_O_2_/HQ) system to process screen-printed carbon electrodes (SPCE) for ampere testing. Two biosensors were proposed, one with anti-5-mC as the bioreceptor to perform sandwich immunoassay, and the other with specific DNA probe and anti-5-mC as bioreceptors to detect methylated DNA. Among them, DNA probes and anti-5-mC sensors are sensitive for detection of bioreceptors and can detect gene-specific methylation. In 2019, Povedano et al. [57] reported that their ctDNA biosensor was able to detect methylation in RNA with a lower limit of 1.25 × 10^−15^ mol/L without changing the modified antibody of the sensor. Compared with traditional methods, the sensor electrode physically adsorbs the antigen, which is simple to operate and low in cost, and can specifically identify ctDNA methylation. Therefore, the sensor strategy is more suitable for applications in the field of real-time detection and mobile health care. As shown in Figure 10, this immunosensor used two different antibodies. The first of them was a 5-mC antibody immobilized on the surface of carboxylic-acid-modified magnetic beads, capable of capturing any ssDNA with ctDNA methylation sequence. The second was an antibody labeled with peroxidase (HRP-anti-ssDNA) as a signaling antibody, recognizing any ssDNA.

## 5. Other ctDNA Biosensing Methods

For the completeness of the discussion on ctDNA biosensors, the optical sensing methods of ctDNA are briefly presented as follows. Chang et al. [58] proposed a fluorescence polarization method to detect ctDNA based on DNA dissociation probe and chain exchange mechanism and used toehold-mediated strand displacement reaction (TSDR) to identify the specific binding of probes to target mutated DNA to generate fluorescence for qualitative analysis. Jin et al. [59] designed and synthesized a water-soluble cationic fluorescent probe and found that fluorescence intensity was linearly proportional to ctDNA concentration at a certain concentration. Chen et al. [60] combined the binding function of peptide nucleic acid (PNA) with the terminal protection function of small molecule linked DNA (TPSMLD) and developed a dual-recognition fluorescent biosensor for the detection of circulating tumor DNA (ctDNA); ctDNA with a low detection limit of 0.3161 pM and good selectivity was obtained by targeting ctDNA with tumor-specific E542K mutation and methylation of PIK3CA gene. Zhai et al. [61] proposed a new ctDNA sensing strategy based on flow cytometry combining enzyme-free amplification and magnetic separation; ctDNA can trigger a hybridization chain reaction to generate a fluorescent long line assembly of DNA and can be further captured by magnetic beads to present fluorescence signals by flow cytometry.

In addition, Pyrak et al. [62] reviewed the most important types of nanosensors based on surface-enhanced Raman spectroscopy (SERS) for DNA detection. This method can also be used to detect ctDNA. SERS takes advantage of the significant increase in the efficiency of Raman signal generation caused by local electric field enhancement near plasmonic (usually gold and silver) nanostructures. Because of its ultra-low detection limit, it is particularly useful in practical analytical applications. According to this method, Zhou et al. [63] proposed a new design idea: construction of SWNT-based SERS assay coupling with RNase HII-assisted amplification for highly sensitive detection of ctDNA in human blood. The limit of detection of the new method was reported to be 3.0 × 10^−16^ mol/L for the KRAS G12DM gene. Conteduca et al. [64] explored the limits of multipath near-field optical trapping and proposed optical trapping of tiny particles, such as DNA fragments, viruses, and vesicles, based on nanostructures. Optical trapping manipulates cells and single molecules by using the force of light. This possibility of capturing large numbers of objects in parallel optics with limited energy and without the need for complex setups will open up new research in many areas of biology and medicine.

Especially important, Li R et al. [65] proposed an amplified colorimetric biosensor for ctDNA. The colorimetric biosensor utilized the amplification property of HCR and high peroxidase with asymmetric catalytic activity to split G-quadruplex DNA zymes, which produces color signals in the presence of ctDNA; this has been successfully applied in complex biological environments such as human blood plasma for ctDNA detection, with the detection limit corresponding to 0.1 pM. Wang et al. [66] developed a rapid centrifugally assisted colorimetric assay using gold nanoparticles (AuNPs) combined with isothermal amplification to detect single nucleotide polymorphisms (SNPs) (G to C mutations) in KRAS and p.G13D in ctDNA. The combination of the two amplification strategies, isothermal amplification and centrifugally assisted assembly, was able to quantitatively and qualitatively distinguish clinically relevant concentrations of approximately 150 nt with a detection limit of 67 pM.

The realization of ctDNA detection based on fluorescent and colorimetric provides valuable theoretical knowledge for ctDNA biosensors in the field of real-time detection and mobile health. Although the detection method based on ctDNA optical reaction can be better used in mobile medicine, its detection limit can only reach pM level; ctDNA detection methods based on electrochemistry have higher sensitivity and detection limits can almost reach the level of fM, as shown in Table 2.

## 6. Future Perspectives

Along with the rise of the family physician industry, the demand for portable, versatile, and reliable medical diagnostic tools have increased, so electrochemical biosensors for ctDNA detection have great potential in point-of-care testing applications. While ctDNA provides new markers for cancer detection, many limitations exist for ctDNA detection by electrochemical biosensors. Current electrochemical biosensors can only detect a known specific biomarker, and clinical diagnosis requires detecting one or more targets in the case of unknown mutations, so ctDNA detection sensors are far from clinical applications and further research is needed. In addition, most sensors have only one detection channel and cannot detect multiple samples at the same time, so the detection throughput is low, and array-based, high-throughput sensors based on microfluidic systems will become the trend. Due to the low abundance of ctDNA in precancerous lesions and early-stage cancer, the sensor is required to have lower detection limits and high specificity. Based on the above research, it can be found that the application of nanomaterials to increase the electrode surface area and conductivity can improve the sensitivity of the electrode sensor significantly. Currently, although the detection of ctDNA by electrochemical sensors has achieved remarkable results, it cannot completely replace traditional tumor imaging technology yet. It is also necessary to combine biosensors with other methods for comprehensive analysis to minimize the damage to the human body during the sampling and diagnostic process.

## 7. Conclusions

This paper reviews the new advances in electrochemical biosensors for the detection of ctDNA. Among the traditional and common clinical technologies for ctDNA detection, PCR and DNA sequencing are mainly used, but these technologies require expensive instruments and complex operations and have low sensitivity and frequent false positives, which are not suitable for point-of-care testing and the Internet of Things. Sensitive and specific biosensors with simpler operation and real-time monitoring will become a powerful tool for the detection of ctDNA. As biosensors can be easily made into portable and low-cost devices, the biosensors will become a trend of the new technological revolution in ctDNA detection and point-of-care testing. This review mainly analyzed the detection of ctDNA by electrochemical biosensors with details on the strategies of sensor modification and signal amplification. This paper followed with an introduction of other biosensing methods for ctDNA detection. At present, the implementation of these methods in the field of precision medicine and mobile health is full of opportunities and challenges, with many challenges to be overcome.

## Figures and Tables

**Figure 1 biosensors-12-00649-f001:**
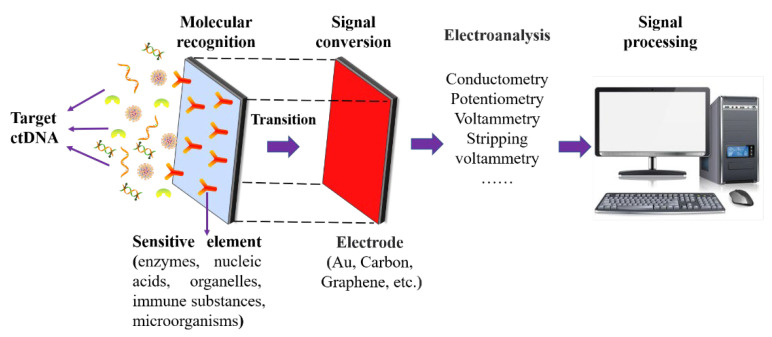
Schematic diagram of ctDNA electrochemical biosensor detection system.

**Figure 2 biosensors-12-00649-f002:**
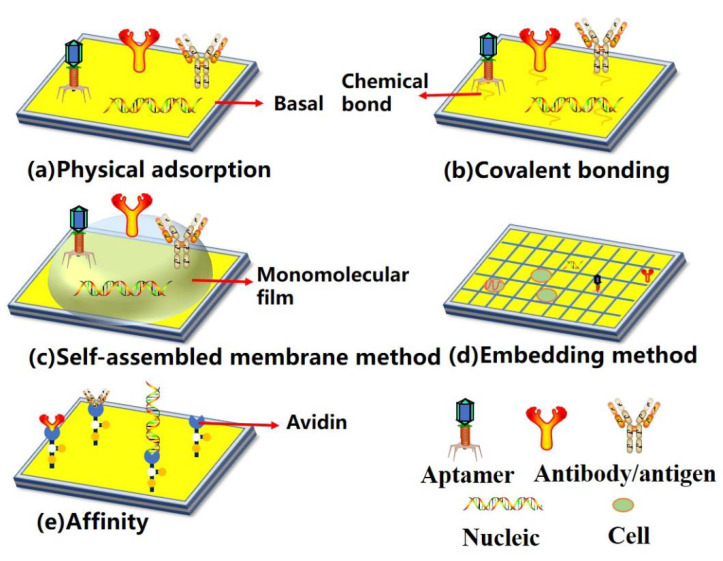
Schematic diagram of sensing surface structure. Biomolecular receptors can be fixed on the surface of a biosensor by (**a**) physical adsorption, (**b**) covalent bonding, (**c**) self-assembled membrane method, (**d**) embedding method, and (**e**) affinity.

**Figure 3 biosensors-12-00649-f003:**
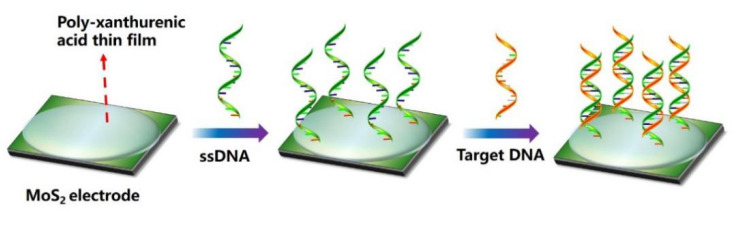
Schematic display of high-performance polymer biosensor based on poly-xanthurenic-acid-functionalized MoS_2_ nanosheets.

**Figure 4 biosensors-12-00649-f004:**
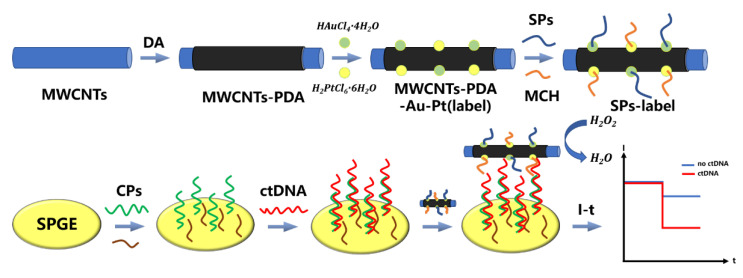
Schematic diagram of sandwich structure ctDNA electrochemical biosensor based on MWCNTs–PDA–Au–Pt nanocomposite and SPs-label (Adapted with permission from Ref. [41]. 2021, Zhao et al).

**Figure 5 biosensors-12-00649-f005:**
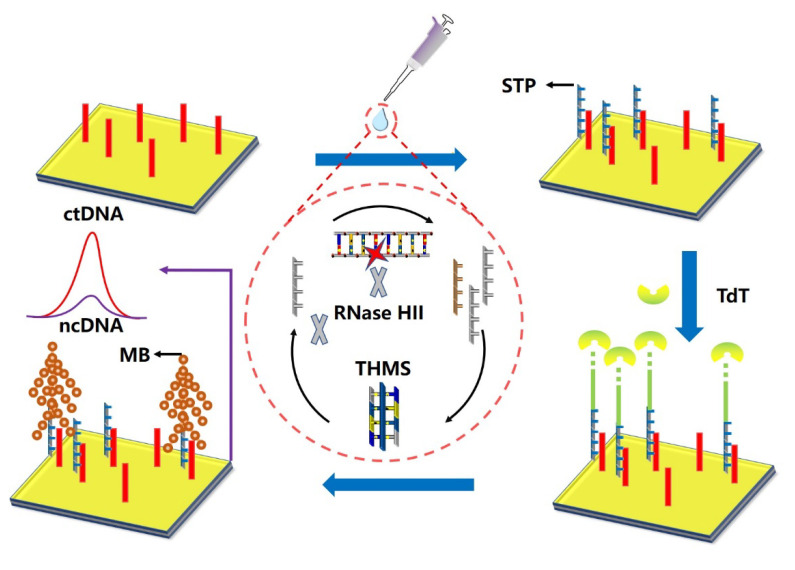
Schematic illustration of the enzyme biosensor using THMS probe and TdT and RNase HII dual amplification (Adapted with permission from Ref. [42]. 2018, Wang et al).

**Figure 6 biosensors-12-00649-f006:**
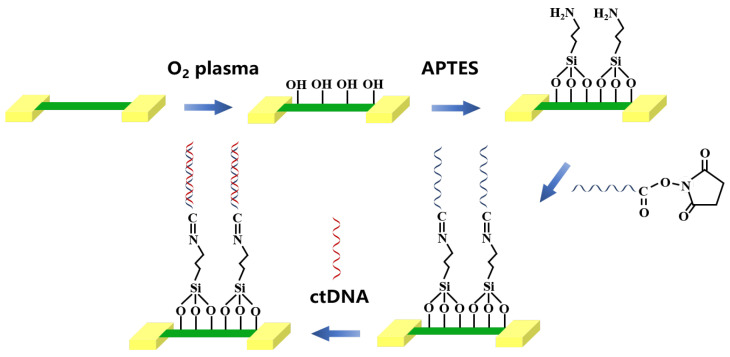
Preparation of SiNW array FET biosensor for tumor marker ctDNA detection (Adapted with permission from Ref. [44]. 2021, Li et al).

**Figure 7 biosensors-12-00649-f007:**
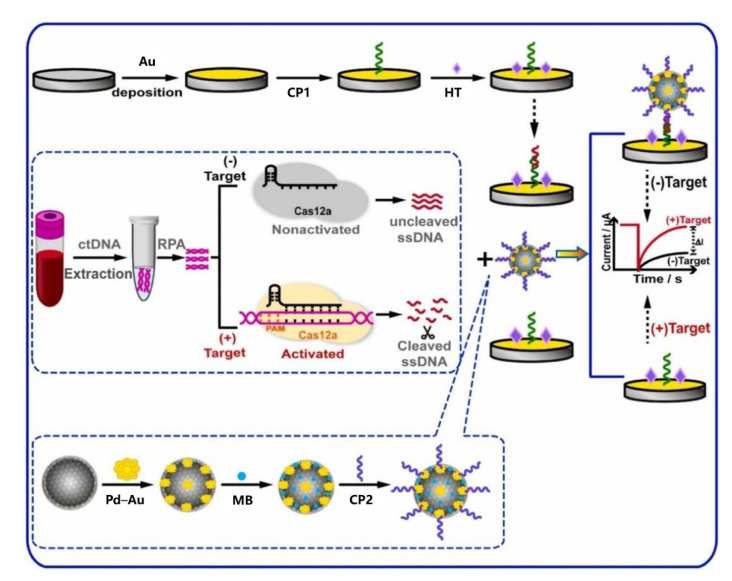
Schematic diagram of ctDNA electrochemical biosensor based on CRISPR/Cas12a (Reprinted with permission from Ref. [46]. 2022, Liu et al).

**Figure 8 biosensors-12-00649-f008:**
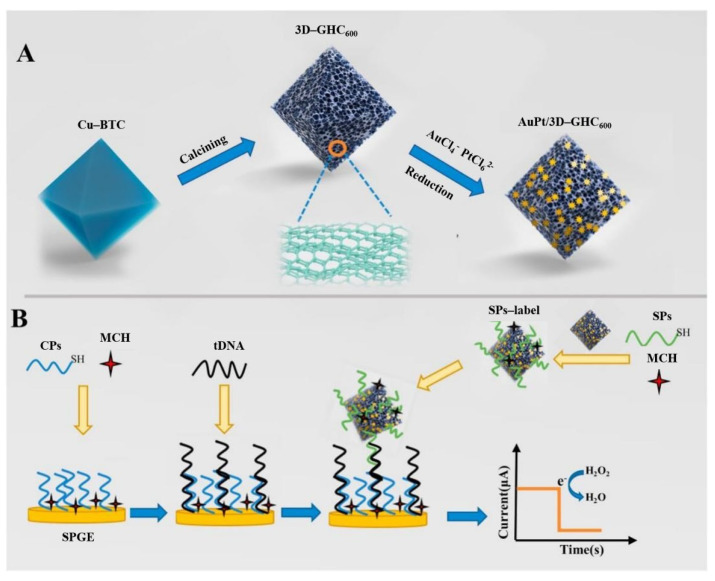
Schematic illustrations of (**A**) the synthetic processes of AuPt/3D-GHC_600_ composite catalyst and (**B**) the fabrication steps of ctDNA electrochemical biosensor (Reprinted with permission from Ref. [47]. 2022, Chen et al).

**Figure 9 biosensors-12-00649-f009:**
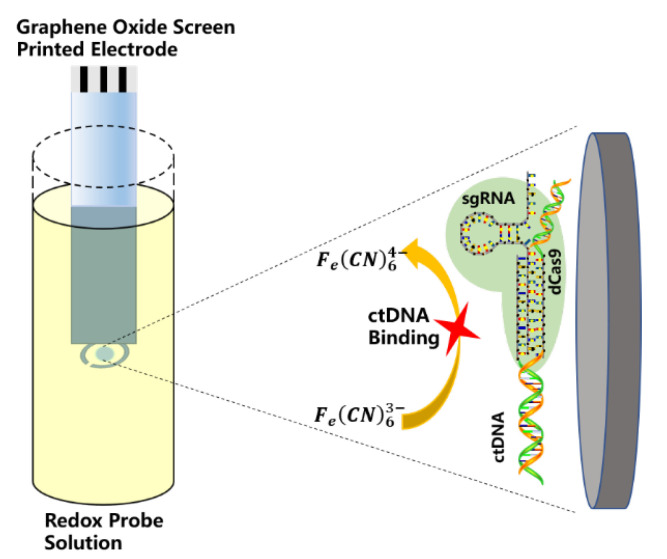
Schematic of a biosensor modified by dCas9–sgRNA (Adapted with permission from Ref. [48]. 2020, Uygun et al.).

**Figure 10 biosensors-12-00649-f010:**
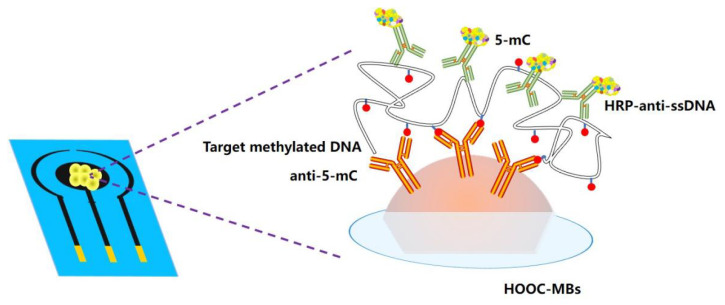
Schematic display of the immunosensor and ctDNA detection (Adapted with permission from Ref. [57]. 2019, Povedano et al.).

**Table 1 biosensors-12-00649-t001:** Biomolecular recognition devices.

Biosensitive Element	Bioactive Unit
Enzymes	Variety of enzymes
Nucleic acids	PNA, DNA, RNA, etc.
Organelles	Mitochondria, chloroplasts
Immune substances	Antigens, antibodies, etc.
Microorganisms	Bacteria, viruses, fungi, etc.

**Table 2 biosensors-12-00649-t002:** Comparison of different methods.

Receptor	Target Species	Electrode	Electrochemical Method	Linear Response Range	LOD	Assay Time	Reference
PNA probe	PIK3CA	SPE	SWV	50–10,000 fM	10 fM	30 min	[34]
	KRASBRAF	Au	DPV	1 fg/μL–100 pg/μL	1 fg/μL	30 min	[37]
	PIK3CA	AuNPs	CV/EIS	50–3200 fM	50 fM	-	[38]
DNA probe	PIK3CA	GCE/rGO-AuNS	CV/EIS/DPV	0.01 aM–1 pM	0.01 aM	-	[39]
	PIK3CA	PXA/MoS_2_/CPE	CV/EIS	0.1 fM–0.1 nM	18 aM	-	[40]
	TNBC	SPGE	CV/EIS	1 fM–10 nM	0.5 aM	-	[41]
	KRAS	Au	DPV	0.01 fM–1 pM	2.4 aM	30 min	[42]
	KRAS	MCH/MB-P1-Fc-P2/U-Au-MGA/GCE	DPV	0.1 fM–1 nM	0.033 fM	60 min	[43]
	PIK3CA	SiNW	I–V characteristics	0.1 fM–100 pM	10 aM	-	[44]
	KRAS	Au	EIS/CV/SWV	10 aM–100 fM	2.2 aM	155 min	[45]
	EGFR	Au/GCE	Chronoamperometry	10 aM–100 pM	3.3 aM	-	[46]
	ctDNA	SPGE	Chronoamperometry	10^−8^ M–10 ^−17^ M	2.25 × 10^−8^ M	-	[47]
RNA probe	PIK3CA	GPHOXE/dCas9-sgRNA	EIS	2–20 nM	0.65 nM	40 s	[48]
Antigen/antibody- based	RASSF1A	SPCE	Amperometric detection	23 pM–24 nM	6.8 pM	45 min	[56]
	RASSF1A	SPCE		139 pM–5 nM	42 pM	60 min	
	5-mC MGMT	SPdCEs	Amperometric detection	4.0–250 pM	1.25 fM	<90 min	[57]
	5-hmC MGMT	SPdCEs		1.44–100 pM	0.43 pM	<90 min	

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
