# Peer review of "Electrochemical Biosensors for Circulating Tumor DNA Detection"

_biosensors, 2022, doi:10.3390/bios12080649_

Round 1

Reviewer 1 Report

I recommend this article for the publication after addressing the bellow-raised quarries and subjected to a minor revision.

1. Abstract needs important information.

2. English needs some improvement (I found some grammatical and typographical errors in the manuscript; author should rectify them).

3. In the introduction part, the information of uses of sensor on the health of living organisms in case of overdose need to be added in the introduction part.

4.  author encourage to include sensitivity of the electrode with comparison table 

5. What is the future perspectives of the sensor 

6. Electrode sensitivity is missing?

7. Recheck all the equations, abbreviations, place of figures and captions.

8. Author should mention the parameters in comparison table 

9. Authors must improvise the citations by citing recent papers like Journal of Science: Advanced Materials and Devices, 2020, 5(4), pp.

Reviewer 2 Report

This article intends to examine the recent developments in biosensors for minimally invasive, real-time, and rapid ctDNA detection. Diverse ctDNA sensors are analyzed based on their choice of receptor probes, electrode designs, detection strategies, and merit scores. I believe that modifications is required to maintain the publication's high quality. 

1.     The authors claim to have discussed the most recent literature, but only one article from the year 2022 and four to five articles from the year 2021 are considered. These are not adequate. 

2.     This amount of review work is insufficient. Additionally, authors should discuss more recent and relevant publications. 

3.     Authors must obtain copyright permission for all reused figures and include a copyright permission statement in the figure's caption. 

4.     Authors should also categorize their work according to the nanomaterials used for sensing. 

5.     Authors should compose the Future Perspectives section separately and in detail. 

6.     In table 1, not every sensor is electrochemical. Please verify and add works to the table. 

7.     The proposed review work must be represented by a solid schematic (add as a Figure 1).

Reviewer 3 Report

The Authors review electroanalytical biosensors for ctDNA detection. The manuscript overviews several devices proposed in literature. Here, my minor comments to the manuscript:

- in order to increase the manuscript appeal, the Authors could describe also the approaches for the samples preparation;

- Innovative techniques have been also proposed in literature for DNA detection, as SERS (Surface enhanced Raman spectroscopy for DNA biosensors—How far are we?. Molecules24(24), 4423, 2019;  Surface-enhanced Raman spectroscopy of DNA. Journal of the American Chemical Society130(16), 5523-5529, 2008) or optical trapping (Exploring the limit of multiplexed near-field optical trapping. Acs Photonics8(7), 2060-2066, 2021; Double nanohole optical tweezers visualize protein p53 suppressing unzipping of single DNA-hairpins. Biomedical optics express5(6), 1886-1894, 2014). Please, discuss also on the them as general discussion.
